# Decoding Strategies to Evade Immunoregulators Galectin-1, -3, and -9 and Their Ligands as Novel Therapeutics in Cancer Immunotherapy

**DOI:** 10.3390/ijms232415554

**Published:** 2022-12-08

**Authors:** Lee Seng Lau, Norhan B. B. Mohammed, Charles J. Dimitroff

**Affiliations:** 1Department of Translational Medicine, Translational Glycobiology Institute at FIU, Herbert Wertheim College of Medicine, Florida International University, Miami, FL 33199, USA; 2Department of Medical Biochemistry, Faculty of Medicine, South Valley University, Qena 83523, Egypt

**Keywords:** galectins, anti-tumor T cell immunity, cell surface glycosylation, cancer therapy

## Abstract

Galectins are a family of ß-galactoside-binding proteins that play a variety of roles in normal physiology. In cancer, their expression levels are typically elevated and often associated with poor prognosis. They are known to fuel a variety of cancer progression pathways through their glycan-binding interactions with cancer, stromal, and immune cell surfaces. Of the 15 galectins in mammals, galectin (Gal)-1, -3, and -9 are particularly notable for their critical roles in tumor immune escape. While these galectins play integral roles in promoting cancer progression, they are also instrumental in regulating the survival, differentiation, and function of anti-tumor T cells that compromise anti-tumor immunity and weaken novel immunotherapies. To this end, there has been a surge in the development of new strategies to inhibit their pro-malignancy characteristics, particularly in reversing tumor immunosuppression through galectin–glycan ligand-targeting methods. This review examines some new approaches to evading Gal-1, -3, and -9–ligand interactions to interfere with their tumor-promoting and immunoregulating activities. Whether using neutralizing antibodies, synthetic peptides, glyco-metabolic modifiers, competitive inhibitors, vaccines, gene editing, exo-glycan modification, or chimeric antigen receptor (CAR)-T cells, these methods offer new hope of synergizing their inhibitory effects with current immunotherapeutic methods and yielding highly effective, durable responses.

## 1. Introduction

Galectins (Gal) are a family of soluble carbohydrate-binding lectins that bind β-galactoside-containing glycans [1]. They are ubiquitously expressed in mammals, amphibians, fish, birds, insects, and fungi [2]. Their expression is largely dependent on the tissue/cell type and the subcellular location [3]. To date, fifteen members of the galectin family have been identified in mammals and classified into three main groups based on their molecular structure (Figure 1A): (i) proto-type galectins (Gal-1, -2, -5, -7, -10, -11, -13, -14, and -15) have a single carbohydrate-recognition domain (CRD) and commonly form homodimers; (ii) tandem-repeat galectins (Gal-4, -6, -8, -9, and -12) have two CRDs connected by a linker peptide; and (iii) chimera-type galectin (Gal-3) has a single CRD linked to an N-terminal tail [4]. Out of the fifteen mammalian galectins, only eleven have been found in humans [5]. Galectins are functionally associated with numerous biological processes, including pre-mRNA splicing, apoptosis, and regulation of the cell cycle, cell adhesion, and signal transduction pathways [6,7]. The functional diversity of galectins is mediated via simultaneous interaction with multiple glycoconjugates using their highly conserved CRDs, and in some cases, through direct interactions with intracellular protein targets [8].

Dysregulated expression of galectins has been reported in a substantial number of studies and frequently correlates with cancer stage and prognosis [8]. Indeed, galectins have been demonstrated to play a crucial role in regulating cancer progression and anti-tumor immune responses [9]. They often contribute to sustained oncogenic signaling, cell death resistance, cell migration, angiogenesis, the evasion of immune surveillance, and chemoresistance [10]. Gal-1, -3, and -9, in particular, have been widely studied in the context of cancer immunology, owing to their prominent roles in fine-tuning T cell survival, differentiation, and anti-tumor function [10]. 

Over the past decade, our understanding of how cancer evades the immune system has deepened, allowing scientists to develop new strategies for halting cancer immune evasion, such as immune checkpoint inhibitors (ICI), adoptive T cell therapy, and vaccines [11,12]. However, efforts are still needed to enhance the efficacy of these immunotherapies and unleash new molecular targets. In this regard, the immunomodulatory role of Gal-1, -3, and -9 has drawn attention to the possibility of targeting them and/or their ligands to overcome any immune evasion shortcomings of current immunotherapies [13,14].

In this review, pertinent information on how Gal-1, -3, and -9 modulate tumor progression will first be introduced. We will then discuss their roles in tuning the anti-tumor immune response and their prospects as novel therapeutic targets to maintain and/or restore anti-tumor T cell function and immunity.

## 2. Molecular Structure of Gal-1, -3, and -9 and their Involvement in Cancer

### 2.1. Gal-1

Gal-1 is a prototype galectin that forms homodimers of non-covalently linked subunits (14 kDa) (Figure 1A) [15]. It was first discovered in 1975 after isolation from an electric eel, and thus, named “electrolectin” [16]. Gal-1 is encoded by the *LGALS1* gene located on chromosome 22q13.1 [17]. Monomeric Gal-1 has six cysteine residues, which render this lectin highly susceptible to oxidation and the subsequent formation of intramolecular disulfide bonds [18]. Gal-1 oxidation induces deep conformational changes that interfere with its dimerization and ligand recognition. Hence, experimentally, the β-galactoside-binding activity of Gal-1 is compromised in the absence of a reducing agent [19].

Gal-1 recognizes glycoconjugates bearing the N-acetyllactosamine (LacNAc) moieties with terminal and internal β1,4-linked galactose residues [20]. Gal-1 binding to LacNAc units is regulated via the action of one of seven ß1,4 galactosyltransferases synthesizing *N*-acetyllactosamine (Gal*β*1-4GlcNAc), and its binding to repeating poly-LacNAc units is regulated via the action of ß1,3 N-acetylglucosaminyltransferase 2 (*B3GNT2*), which is frequently decorated with other functional groups such as sialic acid [21,22,23]. Binding is further enhanced by the amplification of branching in complex-type N-glycans, repeated LacNAc motifs in N- and O-glycans, and 3-*O*-sulfation of the ligand [24]. As such, the action of N-acetylglucosaminyltransferase 5 (*MGAT5*) (which catalyzes the addition of ß1,6-N-acetylglucosamine to the α-linked mannose of biantennary N-glycans) and ß1,6-*N*-acetylglucosaminyltransferase 1 (*GCNT1*) (which create the ß1,6 branch on Core 2 O-glycans [25]) provide added antennae scaffolds for LacNAc formation. On the other hand, Gal-1 binding is inhibited by the expression of β-galactoside α-2,6-sialyltransferase 1 (*ST6GAL1*), which catalyzes the addition of α2,6-linked sialic acids to the termini of N-glycan antennae [26].

A plethora of studies suggest that Gal-1 contributes to tumor growth and progression through its interaction with intracellular and extracellular binding partners [15]. Gal-1 binds to a variety of glycosylated ligands displayed on the surface of immune, stromal, and tumor cells or to extracellular matrix (ECM) proteins [15]. Elevations in Gal-1 in the tumor microenvironment (TME) are well documented in various malignancies [27] and studies reveal key roles of Gal-1–glycosylated ligand interactions in promoting tumor migration and metastasis [28,29,30,31], vasculogenic mimicry [27,32,33], and immunomodulation [34]. Intracellular Gal-1 has also been shown to be involved in several signaling pathways controlling tumor cell proliferation, migration, and invasion in cancer [35,36,37]. For instance, Gal-1 interactions with H-Ras-GTP (the active form of H-Ras) can promote Ras binding to the cell membrane, inducing sustained activation of the Ras/MEK/ERK oncogenic pathway and enhancing cell transformation [38]. Both in vivo and in vitro studies reveal that the upregulation of Gal-1 in ovarian cancer cells promotes tumor cell migration and invasion by activating the MAPK JNK/p38 signaling pathway [39]. In another study, the PI3K-AKT-β-catenin signaling pathway was suppressed in cancer colon cells due to Gal-1 retention at the plasma membrane via its interaction with the intracellular domain of Protocadherin-24 (PCDH24), a non-classical cadherin member that is downregulated in colorectal cancer [40].

### 2.2. Gal-3

Gal-3 was first identified in 1982 as a surface marker on murine peritoneal macrophages, and hence, is known as Mac-2 [41]. Gal-3 is the only member of the chimeric-type group of galectins with a molecular weight of 29–35 kDa [42]. It is encoded by a single gene (*LGALS3*), made up of six exons and five introns, and located on chromosome 14, locus q21-q22 [43]. Gal-3 exists as a monomer in solution with a single CRD and an N-terminal proline- and glycine-rich domain, through which Gal-3 molecules can oligomerize and form pentamers upon glycan binding to the CRDs (Figure 1A) [44].

Gal-3′s preferential glycan ligand is poly-N-acetyllactosamine (poly-LacNAc), a unique glycan structure containing 3–4 repeating units of LacNAc [45]. However, unlike Gal-1, Gal-3-binding ability does not require a terminal β-galactose residue [20]. The N,N′-diacetyllactosamine (GalNAcβ1,4GlcNAc, LacdiNAc), a recently identified epitope in some O- and N-linked glycoproteins, can act as a selective Gal-3 ligand [45]. Interestingly, terminal LacdiNAc expression on glycosylated proteins has been reported to be upregulated in a variety of malignancies, suggesting its potential value as a cancer glycome biomarker [46]. Additionally, sialylation is a major regulator of Gal-3–glycan ligand binding. Gal-3 binding is inhibited by the enzymatic activity of N-acetylgalactosamine α2,6 sialyltransferase 1 (*ST6GALNAC1*), which generates sialyl Tn antigen on O-glycans and prevents LacNAc formation on O-glycans [47]. Gal-3 binding is also inhibited by the action of α2,6 sialyltransferase 1 (*ST6GAL1*) [48].

A considerable amount of research has revealed the prominent role of intracellular and extracellular Gal-3 in tumor growth, progression, and metastasis [49,50]. Intracellular Gal-3 has been reported to mediate the migration of colon cancer cells via activation of the K-Ras–Raf–ERK1/2 signaling pathway [51]. In another study on oral tongue squamous cell carcinoma, Gal-3 has been shown to enhance tumor cell proliferation, migration, and invasion through activation of the Wnt/β-catenin signaling pathway [52]. A more recent study on gastric cancer also reveals that Gal-3 enhances tumor progression by regulating the crosstalk between the WNT and STAT3 signaling pathways [53]. 

Owing to its characteristic oligomerization ability, extracellular Gal-3 plays a crucial role in mediating cancer cell motility by regulating the adhesive interaction of tumor cells with ECM glycosylated ligands, such as fibronectin, laminin, collagen IV, and elastin [54]. Additionally, microenvironmental Gal-3 has been implicated as a key player in establishing the immunosuppressive TME [55]. It has also been reported that Gal-3 regulates capillary tube formation during tumor progression, possibly by binding with αvβ3 integrins on endothelial cells to activate integrin clustering and signaling [56]. 

### 2.3. Gal-9

Gal-9 is a member of the tandem-repeat galectins and is composed of two non-homologous CRDs connected by a peptide linker region at a molecular weight of 34–39 kDa (Figure 1A) [57]. It was first identified in 1997 after isolation from murine embryonic kidney [58] and tumor tissues of patients with Hodgkin’s disease [59]. In humans, Gal-9 is encoded by the *LGALS9* gene, which is located on the long arm of chromosome 17 at locus 11.2 (17q11.2), as well as two LGALS9-like genes (*LGALS9B* and *LGALS9C*) on the short arm of the same chromosome at locus 11.2 (17p11.2) [60]. Gal-9 typically binds internal LacNAc units, with a preference for linear poly-LacNAc glycans [61,62]. Gal-9 can form multivalent lattices due to the different oligosaccharide-binding affinities of its two CRDs [63]. Additionally, its long peptide linker allows the CRDs to have rotational freedom, enhancing multimerization and lattice formation [64]. 

Gal-9 is well known for its immunomodulatory functions, contributing to both innate and adaptive immunity [63]. Gal-9 was first described as a potent eosinophil-specific chemoattractant, derived from activated human T cells to mediate eosinophil adhesion to vascular endothelial cells (ECs) [65]. Recent data also highlight Gal-9′s role as an adhesion molecule bridging neutrophils [66], B cells [67], and CD4^+^ and CD8^+^ T cells [65] to ECs, thus making Gal-9 a crucial regulator of leukocyte trafficking and locomotion. Meanwhile, the role of Gal-9 in cancer progression is controversial and unresolved. Gal-9 can induce apoptosis of hepatocellular carcinoma (HCC), hence inhibiting tumor growth, possibly mediated via the miR-1246-DYRK1A-caspase-9 axis [68]. In another study using highly metastatic melanoma and colon cancer cells, Gal-9 was reported to suppress tumor cell migration and metastasis, both in vivo and in vitro, possibly by blocking the binding of adhesion molecules on tumor cells to their ligands on ECM and vascular ECs [69]. On the other hand, Gal-9 has been found to induce melanoma growth through its binding to CD206 on M2 macrophages, which promotes a tumor-supportive microenvironment [70]. The conflicting role of Gal-9 in cancer biology is thought to be essentially dependent upon its interacting receptor and the particular cancer cell type [71].

## 3. Effects of Gal-1, -3, and -9 on Cancer Immunosurveillance

The process through which the immune system can recognize and eliminate tumor cells is known as tumor immune surveillance [72]. There is immense evidence on how tumor-infiltrated immune cells can regulate tumor development and progression via various extrinsic and intrinsic anti-tumor mechanisms [72]. These immune cell infiltrates include T cells, B cells, natural killer cells, tumor-associated macrophages, and mast cells, with T cells representing the major immune cell subset [73]. However, tumor cells can escape immune destruction through various mechanisms including aberrant tumor antigen processing and presentation [74], enhanced tumor cell resistance to apoptosis [75], the development of immunoinhibitory TME [76], and the induction of tumor antigen-specific T cell tolerance [77]. Importantly, molecular effectors, such as galectins, can mediate tumor immune escape by engaging co-inhibitory receptors, disrupting costimulatory pathways, controlling activation and cytokine secretion, skewing differentiation, and compromising the survival of immune cells [78,79]. In this section, we will discuss how Gal-1, -3, and -9 promote cancer immune evasion (Figure 1B). 

### 3.1. Regulation of T Cell Viability

T cell survival and death are controlled via several regulatory signaling mechanisms [80]. Accumulating evidence suggests that dysregulated expression of Gal-1, -3, and -9 in various types of cancers contributes to the immunosuppressive TME via their binding to T cell surface glycosylated receptors that modulate effector function and, in some cases, induce apoptosis [78]. The mechanisms by which these galectins fine-tune T cell survival are variable, depending on type of galectin, its source, and the glycosylation phenotype of the interacting T cell receptor and related cellular signaling pathway [78,81].

Ongoing research has established that Gal-9 controls T cell survival through its interaction with its receptor, T cell immunoglobulin and mucin domain-containing-3 (TIM-3), to block the T helper type 1 (Th1) cell response and induce apoptosis [82,83]. The importance of the Gal-9–Tim-3 interaction is considered a major immune checkpoint pathway that can be exploited for targeting in immunotherapies. Recently, research by Yang et al. revealed in a murine breast cancer model that an inhibitory receptor, programmed death-1 (PD-1), regulates T cell exhaustion by binding to Gal-9, which inhibits exhausted T cell death and contributes their persistence in tumors [84]. Meanwhile, Gal-1 elicits T cell apoptosis by binding CD3, CD7, CD29, CD43, and CD45 [85,86,87,88,89]. Regarding Gal-1–CD45 interactions, the presence of core 2 *O*-glycans, synthesized by core 2 β1,6-N-acetylglucosaminyltransferase (GCNT1), on the surface of T cells provides Gal-1-binding moieties to induce the clustering of CD45 and cell death [90]. Stillman et al. have shown that, similar to Gal-1, Gal-3 binds to CD29, CD43, and CD45, among which only CD45 interactions seem to contribute to Gal-3-induced cell death. In this respect, CD45 remains diffusely distributed on T cell membranes without clustering [81]. Similarly, studies by Fukumori et al. have reported that Gal-3 binds CD29 (with or without CD7) and mediates Gal-3-induced cell death [91]. Gal-3 can also bind CD71 and induce cell death by mediating CD71 clustering on the T cell surface [81]. Hence, the binding potential of Gal-1, Gal-3, and Gal-9 to various glycosylated T cell receptors can thwart anti-tumor immunity, largely through the promotion of T cell apoptosis. 

The main apoptotic pathways modulated by Gal-1, -3, and -9 include both the mitochondria-associated apoptotic pathway (intrinsic) and the death-ligand/death-receptor pathway (extrinsic) [92]. Other results in Fukumori et al. indicate that extracellular Gal-3 binding to T cell surface glycosylated ligands results in activation of the mitochondrial apoptotic pathway, including cytochrome c release and caspase-3 activation [91]. Unlike cytoplasmic Gal-3, which prefers binding to K-Ras [93], recent findings by Xue et al. showed that extracellular Gal-3 stimulates apoptosis via the activation of extracellular signal-regulated kinase (ERK), independent of Ras and Raf, in addition to the activation of reactive oxygen species and protein kinase C (PKC) [94]. Gal-1-induced T cell apoptosis involves activation of the JNK/c-Jun/AP-1 pathway and the downregulation of anti-apoptotic Bcl-2 expression [95,96]. Gal-1 has also been reported to interact with the extracellular domain of CD95 (APO-1/FAS) on resting T cells, stimulating CD95 clustering and caspase-8 activation to induce T cell apoptosis [92]. In Gal-9-dependent apoptosis, the Ca^2+^-calpain-caspase-1 pathway is implicated in mediating CD4^+^ and CD8^+^ T cell death. Moreover, a repressive signal on activated T cells is induced upon Gal-9–Tim 3 interaction, which induces calcium influx, the triggering of apoptosis, and the suppression of Th1 and Th17 responses [97]. The hijacking of T cell apoptotic pathways triggered by Gal-1, -3, and -9 provides an opportunity to target these events to enhance anti-tumor immunity, an area of interest for future anti-cancer immunotherapeutics. 

### 3.2. Modulation of T Cell Differentiation and Effector Function

Upon TCR recognition of tumor-associated antigens displayed on class I or class II MHC molecules of antigen-presenting cells (APCs), T cells activate, expand, differentiate, and infiltrate the cancer site to eliminate tumor cells [98]. Generally, tumors with high lymphocytic infiltration, particularly CD8^+^ cytotoxic T cells, have been reported to be positively correlated with disease-free survival and/or overall survival in different types of malignancies [99]. However, it is now also considered that tumor-infiltrating immune cells can promote cancer development depending on a combination of factors, such as immune cell type, differentiation or functional state, and the local TME cytokine milieu [73,100]. Additionally, the surface glycome profile of immune cells has been hypothesized to be a key determinant of the immune cell repertoire and behavior in the TME [101,102]. 

Data from Toscano et al. reveal that Th cell subsets display differential glycosylation patterns on their surfaces that impact their susceptibility to Gal-1-mediated apoptosis. Gal-1 can bind and cause apoptosis of pro-inflammatory Th1 and Th17 effector cells, while Th2 cells are protected due to α2,6 sialylation on their surface glycoconjugates, resulting in a tolerogenic microenvironment [101]. Moreover, Gal-1 can generate IL-27-producing tolerogenic dendritic cells (DCs), which drive the differentiation of IL-10^+^ Foxp3^−^ T regulatory type 1 (Tr1) cells, thereby promoting T cell tolerance and dampening Th17 and Th1 responses [103]. Gal-1 also interacts with CD45 on the surface of unpolarized activated Th cells, inducing the development of IL-10-producing Th cells, resulting in a potent anti-inflammatory environment that encourages tumor immune evasion [104]. In classical Hodgkin’s lymphoma, Reed–Sternberg cells express high levels of Gal-1, which enhance the secretion of Th2 cytokines (IL-4, IL-5, IL-10, and IL-13) and promote the expansion of Treg cells [105]. Gal-1 also contributes to the immunosuppressive environment in glioblastoma by upregulating M2 macrophages and myeloid-derived suppressor cells (MDSCs) and inducing the secretion of the immunosuppressive cytokines IL-10 and TGF-β [106]. Consistently, the silencing of glioma-derived-Gal-1 decreases infiltrating macrophages and MDSCs and significantly increases IFN-ɣ secretion by CD8^+^ T cells in tumor-inoculated mice [107]. Similar findings have been observed in pancreatic cancer, demonstrating that Gal-1 deletion represses MDSCs’ development and enhances effector T lymphocyte infiltration [108]. 

Upregulated microenvironmental Gal-3 is also involved in impairing the function of CD8^+^ tumor-infiltrating T lymphocytes (TIL) and thwarting the IFN-γ-chemokine gradient within the tumor matrix. In fact, activated TILs can express surface glycans that have higher binding affinity to Gal-3 compared with resting T cells, facilitating the formation of Gal-3–glycoprotein ligand lattices that interfere with T cell receptor mobility and signaling and hinder IFN-γ diffusion in the TME [109,110]. 

Gal-9–Tim-3 interactions have also been reported to control Th1 immune responses via two mechanisms: directly, as discussed above, by triggering Th1 cell death, and indirectly, by promoting the expansion of immunosuppressive CD11b+ Ly-6G+ MDSC expansion [111]. Additionally, Gal-9 enhances the stability and function of induced Treg cells through its binding to CD44, which forms a complex with TGF-β receptor I, resulting in Smad3-driven transactivation of Foxp3 expression [112]. 

### 3.3. Regulation of Resistance against Cancer Immunotherapies

One of the major obstacles in cancer immunotherapy is the development of therapy resistance, due to the molecular/antigenic evolution of tumor cells and/or dampening of the anti-tumor immune response [113,114]. Accumulating evidence has recently posited that galectins also play key roles in mediating resistance to anti-neoplastic drugs [79]. Data from Lykken et al. reveal that Gal-1 hinders macrophage activation and decreases the sensitivity of non-Hodgkin’s lymphoma to CD20 immunotherapy [115]. In head and neck cancer, tumor-secreted Gal-1 hampers T cell infiltration in the TME by upregulating both programmed death ligand 1 (PD-L1) and Gal-9 and activating the STAT signaling pathway in endothelial cells (ECs), mediating tumor resistance to anti-PD-1 therapy [116]. Recently, publicly available single-cell RNA-seq data of human melanoma have insinuated that TILs from non-responders to therapeutic PD-1 blockade express higher levels of Gal-9 compared to responders. The results also demonstrate that a combined blockade of PD-1 and Gal-9 in tumor-bearing mice results in a higher anti-tumor therapeutic efficacy than either therapy alone [84]. In other work, Gal-9 has been shown to mediate the resistance of pancreatic ductal adenocarcinoma (PDA) cells to chimeric antigen receptor (CAR)-T cell therapy [117]. Additionally, in a clinical study, patients with metastatic melanoma treated with a combination of anti-cytotoxic T lymphocyte-associated antigen-4 (CTLA-4) antibody (Ipilimumab™) and anti-vascular endothelial growth factor A (VEGF-A) antibody (Bevacizumab™) displayed elevations in anti-Gal-3 Abs that correlated with favorable clinical outcomes; this suggested an additional anti-tumor effect directed against Gal-3 via ipilimumab–bevacizumab combination therapy [118]. This study, in fact, demonstrates high pre-treatment levels of circulating Gal-1 and Gal-3, which correlate with poor clinical responses to anti-PD-1 therapy and a lack of post-treatment anti-Gal-1 and -Gal-3 antibodies, implying augmented therapeutic efficacy of Gal-1 and Gal-3 antagonism with anti-PD-1 therapies [118,119]. 

## 4. Strategies to Block Gal-1-, -3, and -9–Ligand Binding as Novel Anti-Cancer Therapies

Novel anti-cancer approaches against galectins are being developed in concert with conventional therapies to overcome therapy resistance, re-invigorate anti-tumor immunity, and, ultimately, establish more durable responses. In this section, we will review anti-cancer approaches that are currently in pre-clinical and clinical trials (Table 1) designed to antagonize Gal-1, -3, and -9–ligand interactions (Figure 1C).

### 4.1. Neutralizing Antibodies

The development of monoclonal antibodies (mAbs) against galectins is a rapidly progressing anti-cancer strategy. Theoretically, these reagents are characterized by a high target specificity, making them great candidates for blocking galectin function in disease settings [138]. Importantly, one of the main consequences of anti-galectin mAbs is the immune-boosting effect that can accentuate the therapeutic efficacy of other immunostimulatory modalities. 

Neutralizing or blocking mAbs for Gal-1 has been reported to inhibit Gal-1–glycan interactions and suppress tumor growth by interfering with Gal-1-dependent pro-angiogenic and immunomodulatory activities. In in vitro and in vivo studies by Stasenko et al., the results revealed that the anti-Gal-3 blockade inhibits AKT and ERK1/2 phosphorylation in MUC16-expressing cancer cell lines, impeding their invasive abilities and prolonging the overall survival of tumor-bearing mice [92]. Similarly, mice inoculated with PDA cells and treated with Gal-9 mAbs showed significant tumor regression via disruption of the Gal-9–dectin1-dependent immunotolerant environment [139].

In addition to direct targeting of galectins, developing mAbs against their respective ligands has demonstrated anti-cancer therapeutic promise. MBG453, also known as Sabatolimab™, is an anti-TIM-3 mAb that interferes with the Gal-9–TIM-3 interaction and has been demonstrated to elicit immune-boosting, anti-leukemic activity [140]. Clinical trials are ongoing to evaluate the safety and efficacy of MBG453 combination therapy in patients with high-risk myelodysplastic syndromes or acute myeloid leukemias and who are not good candidates for intensive chemotherapy [141]. 

### 4.2. Metabolic Modifiers of LacNAc Synthesis and Competitive Inhibitors of Galectin Binding

The inhibition of galectin–ligand interactions can be accomplished via treatment with metabolic modifiers of glycosylation and competitive inhibitors of galectin binding. One approach to disrupting the synthesis of galectin-binding glycans is using metabolic biosynthetic modifiers of glycan biosynthesis, such as fluorinated sugar analogs. By replacing hydroxyl (-OH) groups with a fluorine atom on the hexose/hexosamine ring [142], along with O-acetylation to increase cell permeability, these sugar analogs have shown promise in inhibiting glycan formation through the direct incorporation and truncation of a glycan, the interference of nucleotide sugar formation, and/or the disruption of intracellular levels of nucleotide sugar donors. Peracetylated fluorinated glucosamine analogs, as an example, contain an isosteric substitution of fluorine at the carbon-4 position to block glycosidic bond formation and lessen the production of LacNAcs on activated immune cells [142,143,144]. Our laboratory has demonstrated that tumor growth rates are disrupted when Gal-1-binding glycan ligands on anti-tumor T cells are inhibited by a fluorinated analog of N-acetylglucosamine, 2-acetamido-1,3,6-tri-O-acetyl-4-deoxy-4-fluoro-D-glucopyranose (4-F-GlcNAc) [144]. Other analogs, including fluorinated fucose and sialic acid, also inhibit the formation Lewis X (Le^x^) and sialyl Lewis X (sLe^x^) biosynthesis, which can impair cancer cell adhesion and migration [145]. Similarly, Bull et al. report that sialic acid analogs can block tumor growth by enhancing tumor cell killing by cytotoxic T cells [146]. Of interest, the sugar analog antagonism of immune cell glycan biosynthesis can also benefit patients with autoimmunity. Ex vivo experiments using GlcNAc supplementation on isolated mucosal T cells of the intestinal lamina propria in colonic biopsies and/or peripheral blood mononuclear cells of patients with active ulcerative colitis led to enhancement in the N-glycan branching of T cell receptors; in turn, this led to the suppression of T cell growth, inhibition of the Th1/Th17 immune response, and the control of T cell activity [147]. Thus, the use of sugar analogs needs further exploration to dissociate anti-inflammatory from pro-inflammatory (i.e., anti-tumor) effects that could ultimately control T cell function.

Another method to develop analogs of natural sugar chain precursors and design galectin–ligand inhibitors is quantitative structure–activity relationship (QSAR) analysis. In one recent report, a QSAR has been derived from a library of 136 compounds containing analogs with C3′ and O3′ modifications to help identify a potent, selective inhibitor of Gal-3 binding [148,149]. Interestingly, the results from Kurfirt et al. help identify new multivalent fluorinated sugars, which can abrogate Gal-1 binding, via the deoxyfluorination of -OH residues at carbon positions 3′, 4′, and 6′ of the LacNAc scaffold [150]. 

Other inhibitors of galectin–ligand interactions are represented by neoglycoproteins or non-carbohydrate-based compounds, such as peptides, peptido-mimetics, and heterocyclic compounds [149]. As an example, type-1 LacNAc attached to bovine serum can serve as a scavenger of extracellular Gal-3, thereby protecting T cells from Gal-3-dependent apoptosis [151]. In addition, non-carbohydrate inhibitors for Gal-1 and Gal-3—notably TD139, GM-CT-01, GM-RD-02, MCP, GCS- 100, anginex, 6DBF7, OTX008, and G3-C12—have shown efficacy in inhibiting tumor growth, angiogenesis, proliferation, and the restoration of T cell surveillance in ovarian, melanoma, breast cancer, lung, head and neck, sarcoma, prostate, multiple myeloma, diffuse large B cell lymphoma, and colon cancers [104,149,152]. OTX008 blocks the mitogenic pathway, whereas MCP/GSC-100 activates the caspase-8 and -9 pathways [92]. Since Gal-1, -3, and -9–ligand interactions do play key roles in normal physiologic processes, more pre-clinical safety and anti-tumor efficacy studies are needed for effective translation in humans.

### 4.3. Vaccines

There is a concerted effort devoted to the development of new anti-cancer therapeutic vaccines. Cancer vaccines induce tumor regression by establishing an anti-tumor immune response to tumor cell antigens [153]. Other research strategies in cancer vaccines seek to build protection to pro-malignancy targets, such as Gal-1 and Gal-3. Vaccines using Gal-1 and Gal-3 peptides have shown reductions in tumor growth in pre-clinical mouse models of melanoma and mammary tumor cells [154,155]. A new minigene DNA vaccine encoding Gal-1 epitopes has shown similar enhancement of CD8^+^ T cell immunity with suppression of human neuroblastoma xenografts [156]. The minigene vaccine consists of small DNA-coding sequences in contrast to conventional DNA vectors encoding large full-length sequences that could introduce unsafe genetic material [156]. With early positive results in pre-clinical and clinical studies, galectin vaccines are strategically positioned to grow in promise as their safety and anti-cancer therapeutic efficacy continue to be investigated. With the possibility of generating long-term protection against a cancer-promoting target, cancer vaccines could offer a more economical alternative compared with other short-lived conventional or precision anti-cancer therapeutics. 

### 4.4. Gene Editing

The genetic engineering of glycosylation is an approach, involving the induction or silencing of genes responsible for glycan biosynthesis and degradation, that can potentially be delivered to host or tumor cells and result in potent anti-tumor efficacy [157]. Targeting glycosyltransferase (GT) genes involved synthesizing cancer-associated glycans, for instance, can potentially provide an effective approach to cancer therapy [158,159]. The current approaches used for genetic engineering are the transient overexpression of GTs using cDNA plasmid transfection, RNAi, and modified RNA, and more stable expression approaches via the viral transduction of CRISPR-associated protein 9 (CRISPR/Cas9) knockout vectors or knock-in vectors of specific GTs [157]. Gene editing methods provide spatial and temporal control of specific genes, as well as cell-specific targeting and control, in an inducible manner [160]. For decades, the use of gene editing approaches have allowed researchers to study information on the functional importance of GTs, including galectin–glycan ligand-forming GTs, in cancer, and have been used to model cancer progression and illuminate the pathobiological function and therapeutic promise of glycoconjugate targets in pre-clinical models [31,104,161,162,163,164,165,166,167,168,169,170]. The downregulation of galectins via gene editing methods in combination with chemotherapy has been used to increase therapeutic efficacy against glioblastoma [171]. However, unless gene editing approaches can specifically target the immune, stromal, or cancer cell of interest and largely avoid impacting normal cellular functions, its clinical utility will be severely compromised. Further exploration of a safe and effective approach to altering cancer-promoting features of a target cell glycome is greatly needed.

### 4.5. Exo-Glycosylation

Exciting new studies using extracellular or exo-glycosylation strategies to sculpt cell-surface glycans clearly show effective control over anti-tumor immunity and inflammatory processes by enforcing the functional activity of precise effector immune cell subsets [172,173,174]. The capacity to remodel glycans outside the cell offers a novel approach to manipulating cellular activities and altering physiologic pathways. Sackstein et al. has led this charge by implementing a method known as glycosyltransferase-mediated stereo-substitution (GPS) to enforce the transient expression of E-selectin and L-selectin ligands on the surface of cells [175]. GPS can increase tissue-specific immune cell homing as well as promote the extravasation of multipotent mesenchymal stromal cells (MSCs) to vascular beds expressing E-selectin in vivo [176]. When the expression of hematopoietic cell E- and L-selectin ligand (HCELL), a glycoform of CD44 and the most potent ligand for both E-selectin and L-selectin, is enforced on MSCs via exo-fucosylation, HCELL functions as an MSC bone marrow homing receptor [177,178]. To convert CD44 on MSCs into HCELL, α1,3 fucosyltransferase 6 (FT6) or 7 (FT7) and a nucleotide sugar donor, GDP-fucose, are added in an enzymatic reaction buffer to live cells without affecting viability or multipotency. Recent data demonstrate that the delivery of exo-fucosylated MSCs can also serve as a potent immunomodulatory/anti-inflammatory approach, as the MSC secretome exhibits an increase in anti-inflammatory molecules [175]. Moreover, when comparing FT6 exo-fucosylated MSCs to MSCs transfected with FT6-expressing synthetic chemically modified mRNA (modRNA), FT6-exofucosylation has an immediate peak of sLe^x^ expression followed by a rapid decline within 1–2 days compared to FT6-modRNA, which declines gradually and has a peak ligand expression 2 days after transfection [179]. Thus, these approaches are limited by their exo-glycan expression period, which may compromise sustained homing capacity and long-term tumor regressions.

Regarding other methods, work by Li et al. demonstrates a single-chemoenzymatic-step approach to constructing antibody–cell conjugates [174]. The single-step approach adds macromolecules, such as antibodies, to LacNAc/sialyl-LacNAc moieties on the glycocalyx of cells. A GDP-fucose-containing derivative of the macromolecule is employed via the action of H. pylori α1,3 FT [174]. After the cell has been modified, it exhibits novel functions of specific tumor targeting or resistance to inhibitory signals. For example, when a Herceptin Ab-conjugate was used on human NK-92MI cells (a natural killer cell line) to lyse Her2+ cancer cells, it resulted in enhanced NK cell-killing activity [174]. 

What these techniques and glycan sculpting methods portend is the promise of modifying patient ex vivo-expanded immune cells prior to adoptive transfer. Considering the immune dampening effects of Gal-1, -3, and-9 and their glycan ligands, glyco-engineering patient T cells (or CAR-T cells) with deficits in Gal-binding activity could theoretically improve persistence and effector function in vivo. Utilizing exo-glycoenzymatic modification or the transfection of glycosyltransferase-silencing RNAs or modRNAs encoding glycan-forming enzymes can, perhaps, create a more formidable T cell with a stronger anti-tumor immune response. Efforts involving optimal enzymatic expression/silencing strategies for engineering a more durable anti-tumor adoptive T cell formulation are under development.

## 5. Cell-Based Anti-Cancer Therapies That Can Evade Gal-1, -3, and -9 Immunoregulation 

Cell-based treatments for cancer, in which autologous or allogenic immune cells are isolated and engineered to target cancer cells, represent a highly promising area of cancer therapy development. Adoptive T cell therapy has shown some success in treating the deadly disease metastatic melanoma, with a complete tumor regression rate of 20–25% in clinical trials [180]. Recent phase I/II studies have used interferon-α in combination with patient T cells from PBMC isolates to co-culture with patient tumor cells and expand tumor-reactive T cells that are later reinfused into melanoma patients, resulting in disease control [181]. Maintaining effector anti-tumor T cells within the TME is essential for adoptive anti-tumor T cell therapy. Effective anti-tumor cells are capable of counteracting interference from immunomodulators, such as indoleamine 2,3-dioxygenase (IDO) and Gal-1, -3, and -9, found in TME reservoirs [182]. This idea further highlights the potential of combining methods designed to antagonize galectin–ligand function on ex vivo-expanded anti-tumor T cells, as hypothesized in Section 4.5 above.

Another promising approach is CAR-T cell therapy, in which T cell isolates from PBMC are engineered to express a cell membrane protein that contains an extracellular single-chain variable fragment (scFv) recognition domain; this targets a distinct cancer cell surface antigen and an intracellular signaling module composed of a portion of the T cell receptor (CD)-3 zeta (3ζ) chain to induce T cell activation upon antigen binding [183]. Of note, cell killing exerted by CAR-T cells occurs in an MHC- and Fas-independent manner, which are commonly downregulated in cancer cells to evade immunosurveillance [184,185]. The engagement of the CAR-T cell scFv-tumor antigen causes the degranulation of perforin and granzymes and cytolytic activity [186,187,188]. Once the granzymes enter a target cell, caspase-dependent and -independent apoptotic cell death pathways ensue [189,190]. The design of CAR-T cells has evolved over the years. First-generation CAR constructs consisted only of the CD3ζ domain. Due to the lack of a strong T cell response and limited T cell signaling capability, second-generation receptors were created via the addition of costimulatory signaling domains (e.g., CD28 or 4-1BB), which led to improved cell activation, enhanced survival, and effective expansion of the modified T cells [190,191]. In third-generation CAR-T cell variants, the costimulatory domains CD28 and 4-1BB are now both included in the intracellular costimulatory region. The fourth generation of CAR-T cells or TRUCKs (T cells redirected for universal cytokine-mediated killing) have additional transgenes for cytokine secretion (e.g., IL-12) or additional costimulatory action [192,193,194].

Based on the fundamental molecular mechanism of a CAR, additional modifications are now being developed that enable the targeting of multiple cell types at the same time, as well as CAR with an scFv targeting cancer-associated glycans [183]. He et al. have recently developed a CAR-T cell therapy bispecific for CD13 and TIM-3, both highly expressed on acute myeloid leukemia cells, to improve leukemia cell targeting while sparing normal hematopoietic stem cell killing, and the preclinical in vivo data show striking eradication of leukemic cells in syngeneic and xenograft mouse models [195]. In other exciting work, CAR-T cells engineered to express sLe^X^ via GPS exo-fucosylation with FT6 resulted in improved CAR-T cell homing to the bone marrow [172]. It is hypothesized that improved homing can allow for efficient localization to the tumor site with the administration of fewer cells and can significantly reduce the burden of cell expansion, thus yielding more efficient CAR-T cell therapeutics with higher clinical efficacy and less neurotoxicity [172,173]. Moreover, current research is exploring cancer-associated glycans in solid tumors, such as Le^y^, TAG72, disialoganglioside GD2, and Tn-MUC1, due to their aberrant expression as targets for CAR-T cell therapy [183]. Many of these anti-glycan CAR-T cells have been shown to be successful and are currently undergoing clinical trials. Thus, glycoscience-based CAR-T cell formulations—either via the exo-glycosylation crafting of CAR-T cells for more efficient trnsport to tumors or via the design of CAR-T cells to target cancer-associated glycans—are gaining attention in the cell-based anti-cancer immunotherapy field. 

A clear prospect in this surge of glycobiological methods to improve the anti-cancer efficacy of cell-based immunotherapies is targeting the functional context of Gal-1, -3, and -9–ligand interactions. As cancer glycobiologists continue to explore and identify the key glycan structures and the glycan-forming/inhibiting enzymatic machinery of Gal-1, -3, and -9 ligands, there is momentous potential in designing ex vivo-expanded patient T cells, including CAR-T cells, to evade Gal-1, -3, and -9 binding and related immunomodulation. The ablation of the Gal-binding glycan moieties (e.g., diminish LacNAc levels) or building of Gal-inhibitory moieties (e.g., α2,6 sialylation) through exo-glycosylation or gene editing methods (described in Section 4) could potentially generate a cell designed to evade Gal binding, improve the effector function profile (e.g., increase inflammatory vs. regulatory cytokines), and resist pro-apoptotic induction, resulting in a longer functional half-life. As a result, following adoptive T cell transfer into patients, there is the promise of yielding a stronger, more sustained anti-cancer immune response. Such strategies are under intense investigation with the hope of effectively avoiding the immunocompromising effects of elevated serum levels of Gal-1, -3, and/or -9 associated with several different cancer-types. 

## 6. Conclusions and Future Perspectives

In this review, we describe the emerging roles of Gal-1, -3, and -9 and their plethora of immunosuppressive functions in cancer progression that support the argument for their potential as therapeutic targets in combination with other cancer treatments. Intense efforts seeking to understand how Gal-1, -3, and -9 support cancer growth, dissemination and persistence in distant tissues, in addition to their role in resistance to current cancer immunotherapies, enable exciting new clinical trials that target their action while combining them with other promising cancer therapies [10]. Depending on the galectin expression level in/on distinct T cell subsets or their levels in the TME, their effector anti-tumor T cell viability or cytolytic and pro-inflammatory cytokine secretion capacity can be impacted [15]. Other signature glycan features and their presence on select T cell surface receptors help confer Gal-1, -3, or -9 immunomodulatory effects, providing additional opportunities to target the galectin–ligand immune-modifying axis. 

Over the last decade, there has been a growing number of reports identifying anti-cancer therapeutic strategies that interfere with Gal-1, -3, and -9–ligand interactions. Current inhibitors that target galectin–ligand activity can help restore and/or maintain effector anti-tumor T cell function. The preclinical and clinical data on Gal-1, -3, and -9 neutralizing antibodies, inhibitors, vaccines, gene editing, and exo-glycosylation, above all, serve to boost anti-tumor activity, either as a single agent or in combination with current immunotherapeutic approaches. There are now tangible galectin inhibitor approaches that are undergoing clinical trials in combination with other anti-cancer therapeutics, as listed in Table 1.

To best study and improve anti-galectin–ligand approaches, there needs to be a deliberate effort to use human translational models, as human glycobiology does not structurally mirror animal cell models or murine syngeneic/genetic tumor model systems. Furthermore, because immunoreactivity and immune protection mechanisms can, potentially, be affected by aberrant (therapeutically induced) galectin–ligand levels, side-effect safety profiling needs to be emphasized. Elucidating the proper dosing, multi-intervention scheduling, and ex vivo analyses of galectin–ligand targeting in tumor and immune tissues also necessitate a collaborative research effort between translational glycoscientists and clinical investigators. As this exciting new research evolves, it may be possible to determine its purpose in treating multiple cancer types, along with the deadly consequences of metastatic disease.

## Figures and Tables

**Figure 1 ijms-23-15554-f001:**
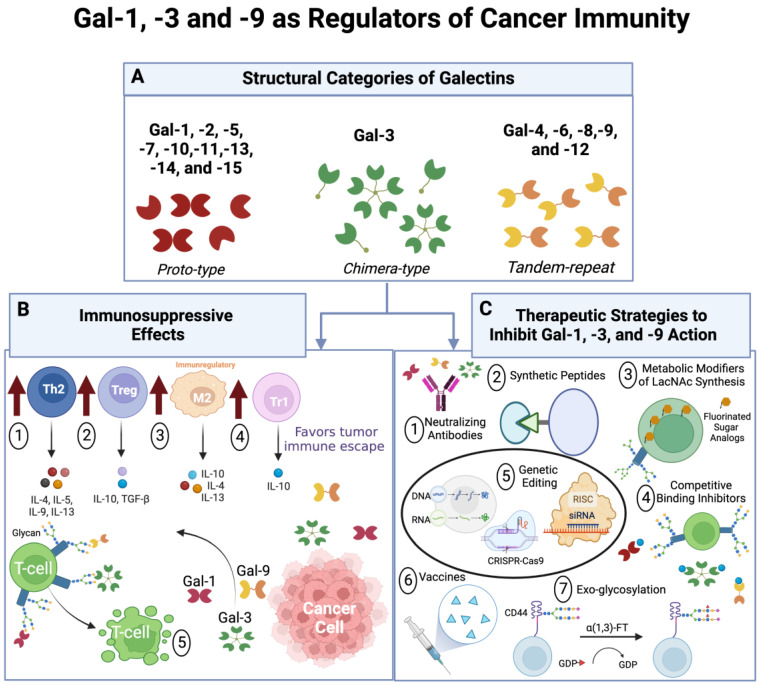
**Gal-1, -3, and -9 as regulators of cancer immunity**. (**A**) Structural categories of galectins: *proto-type*, containing one carbohydrate-recognition domain (CRD); *chimera-type*, with one CRD and an amino-terminal polypeptide tail region that forms pentamers in serum; and *tandem repeat-type* galectins, which contain two distinct CRDs. (**B**) Immunosuppressive effects: **(1.)** Th2 cell skewing; **(2.)** Treg cell expansion and suppressive activity; **(3.)** elevation in M2 macrophages; **(4.)** production of IL-10-producing T regulatory (Tr1) cells; and **(5.)** apoptosis of effector anti-tumor T cells. (**C**) Therapeutic strategies to inhibit Gal-1, -3, and -9 Action. Diagram was created using BioRender.com.

**Table 1 ijms-23-15554-t001:** Targeting Gal-1, -3, and -9–ligand axis in combination therapies currently in clinical trials.

Galectin	Type of Inhibitor	Molecule	Cancer Model	Clinical Trial	Combination Therapy	Citation
Gal-1/Gal-3	Allosteric. inhibitor	GM-CT-01 (Davanat^®^)	Colorectal, lung, breast, head and neck, and prostate cancers	NCT00054977	5-fluorouracil (Adrucil^®^)	[120]
Gal-3	Allosteric inhibitor	GM-CT-01 (Davanat^®^)	Cancers of the bile duct and gallbladder	NCT00386516	5-fluorouracil (Adrucil^®^), folinic acid (Leucovorin), bevacizumab (Avastin^®^)	[121]
Gal-3	Allosteric inhibitor	GM-CT-01 (Davanat^®^)	Colorectal cancer	NCT00388700	5-fluorouracil, folinic acid (Leucovorin), bevacizumab (Avastin^®^)	[122]
Gal-3	Allosteric inhibitor	GM-CT-01 (Davanat^®^)	Metastatic melanoma	NCT01723813	Tumor-specific peptide: MAGE-3. A1/or NA17.A2	[123]
Gal-3	Competitive inhibitor	GCS-100 Modified citrus pectin (MCP)	Diffuse large B-cell lymphoma	NCT00776802	Etoposide (Vepesid^®^); dexamethasone (Maxidex)	[124]
Gal-3	Competitive inhibitor	GCS-100 MCP	Chronic lymphocytic leukemia	NCT00514696		[125]
Gal-3	Competitive inhibitor	GCS-100 MCP	Multiple myeloma	NCT00609817	Bortezomib (Velcade^TM^)/ dexamethasone (Maxidex)	[126]
Gal-3	Competitive inhibitor	GR-MD-02 (Belapectin)	Metastatic melanoma, head and neck squamous cell carcinoma	NCT04987996	Pembrolizumab (Keytruda^®^)	[127]
Gal-3	Competitive inhibitor	GR-MD-02 (Belapectin)	Melanoma, non-small-cell lung cancer, and squamous cell head and neck cancer	NCT02575404	Pembrolizumab (Keytruda^®^)	[128]
Gal-3	Allosteric inhibitor	GB1211	Non-small-cell lung cancer	NCT05240131	Atezolizumab (Tecentriq^®^)	[129]
Gal-3	Neutralizing antibody	MBG453	Myelodysplastic syndrome or chronic myelomonocytic leukemia-2	NCT04266301	Azacitidine (Vidaza^TM^)	[130]
Gal-3	Neutralizing antibody	MBG453	Acute myeloid leukemia	NCT04150029	Azacitidine (Onureg^®^) and venetoclax (Venclexta^®^)	[131]
Gal-3	Competitive inhibitor	GR-MD-02 (Belapectin)	Metastatic melanoma	NCT02117362	Ipilimumab (Yervoy^®^)	[132]
Gal-3	Competitive inhibitor	Modified Citrus Pectin (MCP, PectaSol-C)	Prostate cancer	NCT01681823	5-fluorouracil (Adrucil^®^)	[133]
Gal-9	Neutralizing antibody	LYT-200	Cholangiocarcinoma, colorectal cancer, pancreatic cancer	NCT04666688	Anti-PD-1 mAb, gemcitabine (Gemzar^TM^)/ nab-paclitaxel	[134]
Gal-9/Tim-3	Neutralizing antibody	MBG453 *(Sabatolimab*)	Colon cancer and melanoma	NCT02608268	PDR001 *(Spartalizumab)*	[135]
Gal-9/Tim-3	Neutralizing antibody	MBG453 *(Sabatolimab*)	Acute myeloid leukemia	NCT04150029	Azacitidine (Onureg^®^) and venetoclax	[131]
Gal-9/Tim-3	Neutralizing antibody	MBG453 *(Sabatolimab*)	Myeloid leukemia, acute myelodysplastic syndromes, preleukemia, myelomonocytic leukemia	NCT03066648	Decitabine (*Dacogen^TM^*) or azacitidine (Onureg^®^)	[136]
Gal-9/Tim-3	Neutralizing antibody	MBG453 *(Sabatolimab*)	Glioblastoma multiforme	NCT03961971	-	[137]
Gal-9/Tim-3	Neutralizing antibody	MBG453 *(Sabatolimab*)	Myelodysplastic syndrome or chronic myelomonocytic leukemia-2	NCT04266301	Azacitidine (Onureg^®^)	[130]

## Data Availability

Not applicable.

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
