# Peer review of "Decoding Strategies to Evade Immunoregulators Galectin-1, -3, and -9 and Their Ligands as Novel Therapeutics in Cancer Immunotherapy"

_ijms, 2022, doi:10.3390/ijms232415554_

Round 1

Reviewer 1 Report

The review by Lau et al. provides a comprehensive view of the role of some galectins and other glyco-molecules in cancer immune escape. I have only minor remarks, detailed below.

The Title does not reflect precisely the content of the review, which extends beyond galectins. I think it should be modified accordingly.

Make uniform glycosyltransferase names. For example, ST6GAL1 has been properly named in line 102 but ST6Gal-1 in line 143. I strongly recommend to use for glycosyltransferases the gene name in capital letters without hyphen. Thus, not ST6GalNAc-1 (line 141) but ST6GALNAC1.

Enzyme GCNT1 is introduced for the first time in lines 98-100 and again in lines 229-230 in which a second non-standard abbreviation (C2GnT) is used. Please make uniform.

Check spelling in line 225, 231, 534.  

The chapter 4.5 “Exo-glycosylation” is very interesting but it does not deal with galectins. In addition, some parts should be explained better. In particular, the text between lines 487-495 should be made clearer.

What is modRNA (lines 482 and 483)?  

The text in lines 508-512 should be re-written in a clearer form.

Line 516. The intracellular portion of the CAR is not “an antigen recognition domain”. Rather, it is a signal transduction domain.

Section 5 deals mainly with non-galectin targets.

The legend of Fig. 1C is redundant. The different points (1. neutralizing antibodies ecc.) don’t need to be repeated in the legend, since they are well explained in the Figure.

Author Response

Dear Reviewer#1,

Please see attached Rebuttal Letter.

Thank you!

Reviewer 2 Report

Dear authors, Thank you very much for this interesting review. 1. In the line 49, you have mentioned Gal 15, I assume you wanted to include it in figure A but it was forgotten. 2. In the line 538, I believe HSC stands for hematopoietic stem cells. 3. I highly recommend including illustartions for the paragraphs (2.1. Gal-1) and  (2.2. Gal-3) to be more facile. Best wishes

Author Response

Dear Reviewer#2,

Please see attached Rebuttal Letter.

Thank you!
